# The Promise of Piperine in Cancer Chemoprevention

**DOI:** 10.3390/cancers15225488

**Published:** 2023-11-20

**Authors:** Salma Benayad, Hicham Wahnou, Riad El Kebbaj, Bertrand Liagre, Vincent Sol, Mounia Oudghiri, El Madani Saad, Raphaël Emmanuel Duval, Youness Limami

**Affiliations:** 1Laboratory of Health Sciences and Technologies, Higher Institute of Health Sciences, Hassan First University of Settat, Settat 26000, Morocco; salmabenayad6@gmail.com (S.B.); elkebbajriad@gmail.com (R.E.K.); saad.elmadani@uhp.ac.ma (E.M.S.); 2Laboratory of Immunology and Biodiversity, Faculty of Sciences Ain Chock, Hassan II University, Casablanca 20100, Morocco; hwwahnou@gmail.com (H.W.); mounia.oudghiri@univh2c.ma (M.O.); 3Le Laboratoire des Agroressources, Biomolécules et Chimie pour l’Innovation en Santé (LABCiS), University Limoges, UR 22722, F-87000 Limoges, France; bertrand.liagre@unilim.fr (B.L.); vincent.sol@unilim.fr (V.S.); 4The Franch Center for Scientific Research (CNRS), Université de Lorraine, L2CM, F-54000 Nancy, France

**Keywords:** piperine, cancer chemoprevention, signaling pathways, TRAIL-based therapy

## Abstract

**Simple Summary:**

With 19.3 million new cancer cases worldwide and nearly 10.0 million cancer-related deaths cancer is a global health challenge. Cancer is a disease caused by a variety of factors that lead to uncontrolled cell growth. Cancer chemoprevention using natural or synthetic compounds is a critical strategy to mitigate the impact of cancer on healthcare systems. One of the naturally occurring compounds, piperine, has shown potential in preventing cancer. Piperine is a nitrogen-containing alkaloid molecule with pleiotropic therapeutic effects, including antioxidant, anti-inflammatory, and immunomodulatory properties. With its various health benefits, piperine shows promise in inhibiting cancer-related molecular processes and enhancing existing treatments. This review explores piperine’s potential in cancer prevention and treatment, suggesting it could be a valuable tool in managing cancer.

**Abstract:**

Cancer, characterized by the unregulated growth and dissemination of malignantly transformed cells, presents a significant global health challenge. The multistage process of cancer development involves intricate biochemical and genetic alterations within target cells. Cancer chemoprevention has emerged as a vital strategy to address this complex issue to mitigate cancer’s impact on healthcare systems. This approach leverages pharmacologically active agents to block, suppress, prevent, or reverse invasive cancer development. Among these agents, piperine, an active alkaloid with a wide range of therapeutic properties, including antioxidant, anti-inflammatory, and immunomodulatory effects, has garnered attention for its potential in cancer prevention and treatment. This comprehensive review explores piperine’s multifaceted role in inhibiting the molecular events and signaling pathways associated with various stages of cancer development, shedding light on its promising prospects as a versatile tool in cancer chemoprevention. Furthermore, the review will also delve into how piperine enhances the effectiveness of conventional treatments such as UV-phototherapy and TRAIL-based therapy, potentially synergizing with existing therapeutic modalities to provide more robust cancer management strategies. Finally, a crucial perspective of the long-term safety and potential side effects of piperine-based therapies and the need for clinical trials is also discussed.

## 1. Introduction

Cancer is a global crisis responsible for a staggering number of deaths worldwide. The statistics have painted a somber picture in recent years, with millions succumbing to this relentless disease. In 2020, there were approximately 19.3 million new cancer cases worldwide, excluding nonmelanoma skin cancer, and nearly 10.0 million cancer-related deaths, also excluding nonmelanoma skin cancer. Female breast cancer emerged as the most diagnosed cancer, with approximately 2.3 million new cases, followed by lung (11.4%), colorectal (10.0%), prostate (7.3%), and stomach (5.6%) cancers. However, lung cancer remained the leading cause of cancer-related deaths, with an estimated 1.8 million fatalities (18%), followed by colorectal (9.4%), liver (8.3%), stomach (7.7%), and female breast (6.9%) cancers. Notably, there was a significant difference in cancer incidence between transitioned and transitioning countries, with incidence rates being two to three times higher in transitioned countries for both men and women. Moreover, mortality rates varied less, particularly among women [1,2].

Researchers have focused on chemopreventive natural or synthetic compounds in pursuing improved cancer therapies, uncovering promising candidates [3,4,5,6,7,8,9].

Among these natural compounds, piperine has attracted significant interest due to its diverse mechanisms of action. Piperine, chemically identified as 1-[5-[1,3-benzodioxol-5-yl]-1-oxo-2,4-pentadienyl] piperidine, represents a nitrogen-containing alkaloid molecule with a rich historical background. Its initial discovery traces back to 1820 when the Danish chemist Hans Christian Orstedt first isolated it from the dried fruit extract of pepper, presenting itself as a yellow crystalline solid (with a molecular weight of 285.33 g/mol and a melting point of 128–130 °C) [10]. Piperine’s chemical structure comprises conjugated aliphatic chains that serve as a bridge between the piperidine and 5-(3,4-methylenedioxyphenyl) moiety [11] (Figure 1).

Its pleotropic effects, from mitigating inflammation and triggering cell apoptosis to suppressing cancer stem cells and disrupting the cell cycle, underscores its potential significance in modulating cancer development and progression [12,13,14,15,16,17] (Figure 2). Piperine’s unique property of selectively targeting malignant cells while sparing healthy ones positions it as a compelling contender for cancer therapy, addressing the challenge of distinguishing cancerous cells from healthy ones often faced in conventional treatments [18].

Furthermore, piperine can complement conventional cancer treatments by enhancing their efficacy, reducing adverse effects, and improving drug bioavailability [19]. Its capacity to sensitize cancer cells to radiation therapy, surmount multidrug resistance, and synergize with treatment approaches such as TRAIL-based therapy highlights its versatility in the battle against cancer [20,21,22].

In this comprehensive review, we navigate the intricate landscape of piperine’s contributions to cancer prevention and therapy targeting various cancers including breast, cervical, prostate, lung, skin, stomach, liver, colorectal, and bone cancers, in both in vitro and in vivo models (Table 1). By unraveling its diverse mechanisms and therapeutic potential, we aim to illuminate the promising role of piperine in the ongoing battle against cancer.

To ensure the precision and comprehensiveness of our review, our team undertook a meticulous data collection and search procedure encompassing a diverse array of reputable databases, which included Google Scholar, Pubmed, Springer, Elsevier ScienceDirect, and Web of Science. Our focus was primarily on scrutinizing studies published between 2002 and 2023, with the inclusion of two earlier studies for contextual purposes.

To maintain uniformity and guarantee a thorough analysis, we exclusively selected articles containing English text. We conducted keyword and heading searches for the following terms: “piperine”, “cancer chemoprevention”, “cell death”, “antitumor”, “in vivo studies”, “in vitro studies”, and “combination cancer therapies”. Subsequently, we conducted a rigorous selection process. Prior to proceeding with a detailed examination of full-text documents, we eliminated duplicate papers and irrelevant works. Our inclusion criteria were stringent and encompassed original articles and review papers that met specific criteria, thereby ensuring the precision and quality of the information presented in this paper. In total, 96 references were selected for inclusion in our study.

## 2. Piperine’s Dual Mechanisms of Prevention and Destruction of Cancer

Cancer chemoprevention emboldens the use of natural and synthetic biologically active substances to prevent, inhibit, or reverse cancer progression. Chemopreventive agents have been classified into blocking agents and suppressing agents. Blocking agents impede the initiation of tumors [23,24]. Suppressing agents, on the other hand, act subsequently by suppressing the transformation of initiated cells into preneoplastic and/or neoplastic cells and malignancy [24] (Figure 2). Piperine exhibits a unique duality in its abilities, functioning as both a blocking and a suppressing agent in cancer prevention and therapy. This dual role allows piperine to target multiple pathways and aspects of cancer development and progression, ultimately enhancing the effectiveness of chemoprevention strategies (Table 1).

In this section, we will discuss the detailed mechanistic role of piperine in cancer chemoprevention and destruction.

### 2.1. Piperine Reduces Inflammation

Inflammation has become a target for cancer prevention and therapy. Many studies have evaluated the anti-inflammatory activity of piperine. In fact, in vitro studies using a variety of isolated cells have been used to screen the anti-inflammatory activity of piperine [12,25]. In a model of LPS-induced inflammation of nucleus pulposus cells, piperine significantly inhibited multiple inflammatory factors and oxidative stress-associated genes [26]. In human peripheral blood mononuclear cells (PBMCs), piperine inhibits IL-2 and interferon-gamma (IFN-γ) production [27]. Additionally, in a model of oxidative stress induced by UV-B in HaCaT keratinocyte cells, piperine also inhibits ROS/RNS production, leading to a decrease in inflammatory mediators such as p38, JNK, AP-1, iNOS (nitric oxide synthase), and COX-2 (cyclooxygenase-2) protein expression [12]. Piperine also inhibits LPS-induced tumor necrosis factor (TNF)-α, IL-6, IL-1β, and prostaglandin E_2_ (PGE_2_) production in BV2 microglial cells [28]. Furthermore, piperine down-regulates pathways associated with IL-1β and nuclear factor-κB (NF-κB) [29], activates the Nrf2/keap1 pathway, and inhibits TNF-α-induced expressions of cell adhesion molecules such as ICAM-1, VCAM1, and E-selectin. This leads to the blocking of neutrophil adhesion to the endothelium in a time- and concentration-dependent manner [13,14]. Similarly, in vivo studies using a wide range of acute and chronic experimental models revealed the anti-inflammatory activity of piperine in a dose-dependent way. In experiments involving carrageenan-induced paw edema, formalin-induced arthritis, croton oil-induced granuloma pouch, and cotton pellet-induced granuloma, piperine demonstrated a 56%, 40%, 40%, and 10% inhibition of inflammation, respectively [30,31]. In a research study conducted on fibroblast-like synoviocytes stimulated with IL-1β, derived from patients having rheumatoid arthritis and in an animal model of arthritis, it was found that piperine inhibits IL-6, MMP-13, and AP-1. It also reduces PGE_2_ levels in a dose-dependent manner. Furthermore, piperine-treated rats showed a significant reduction in nociceptive and arthritic symptoms [32].

In summary, piperine consistently proves its potent anti-inflammatory properties across diverse models. Its efficacy is evident in inhibiting inflammatory factors, modulating cytokine production, and down-regulating key pathways. The dose-dependent inhibition of inflammation in various experimental models underscores piperine’s robust potential for addressing inflammatory-related conditions in future research.

### 2.2. Piperine Induces Various Cell Death Types

In this section, we discuss how piperine induces various cell death types, including apoptosis, autophagy, ferroptosis, and anoikis, which play significant roles in cancer prevention and therapy.

#### 2.2.1. Apoptosis

Apoptosis can be caused by two main pathways: the intrinsic and extrinsic pathways [33]. Apoptosis can be triggered by different factors and the way the signal is transmitted also varies, but ultimately, they converge onto the same execution pathway. This results in several changes, such as DNA fragmentation, cytoskeletal and nuclear protein degradation, protein cross-linking, the formation of apoptotic bodies, the expression of phagocytic cell receptor ligands, and finally, uptake by phagocytic cells [34].

Apoptosis is known to be induced by many chemopreventive agents and piperine can activate both the intrinsic and extrinsic pathways of apoptosis. A study performed by Tawani et al. reported the cytotoxic effects of piperine against four different human cancer cell lines (i.e., breast carcinoma (MCF-7), liver carcinoma (HepG2), cervical carcinoma (HeLa), and prostate cancer cells (PC3)), showing apoptotic characteristics such as cytoplasmic and nuclear condensation, where the externalization of membrane phospholipid phosphatidylserine and DNA cleavage were also observed [32]. Furthermore, piperine has been shown to target human G-quadruplex DNA sequences, a structure playing a vital role in regulating cellular processes that might contribute to cancer development [32]. These results were confirmed by Jafri et al., showing that piperine induces apoptosis in a dose-dependent manner by increasing reactive oxygen species (ROS) generation, nuclear condensation, the disruption of mitochondrial membrane potential, DNA fragmentation, and finally, the activation of caspase-3 [35] (Figure 3). Furthermore, in vivo studies showed that piperine reduced tumors in an osteosarcoma xenograft mouse model by up-regulating both Bax and p53 expression, as well as reducing Bcl-2 expression [36].

#### 2.2.2. Autophagy

Autophagy is also an interesting way by which some chemopreventive agents could act. It is a fundamental cellular process that eliminates molecules and subcellular elements via lysosome-mediated degradation to promote homeostasis, differentiation, development, and survival [37]. In the context of cancer, autophagy prevents tissue damage and cell death, which can cause cancer initiation and progression [38]. Furthermore, it is known that the central pathway governing autophagy is led by PI3K/Akt/mTOR signaling [32,39]. Strikingly, piperine has been shown to induce autophagy in cancer cells by inhibiting mTORC kinase activity, allowing the formation of autophagosomes [40,41]. In this context, piperine inhibited thioredoxin reductase (TrxR) activity, increased ROS levels, reduced mitochondrial membrane potential, and induced autophagy in Bel-7402/5-FU cells via the regulation of autophagy-related proteins LC3, p62, and beclin-1 [15] (Figure 3). In vivo experiments demonstrated that piperine induces autophagy by inhibiting PI3K signaling, leading to a decrease in oral cancer tumor growth [42].

#### 2.2.3. Ferroptosis

Ferroptosis is also an interesting target of chemopreventive agents. Ferroptosis is characterized by iron and lipidic ROS/peroxides accumulation due to the Fenton reaction and via the loss of balance in ROS production and cell glutathione (GSH)-dependent antioxidants, which protect cells from lipid peroxidation [43]. Piperine could increase the intracellular Ca^2+^ level and ROS in cancer cells also activate the Fenton reaction at high concentrations [24,43,44,45] (Figure 3).

#### 2.2.4. Anoikis

Anoikis, a particular programmed cell death, could be induced by a piperine structural analogous named piperlongumine in melanoma cells in vitro by inhibiting STAT3 [46] (Figure 3).

### 2.3. Piperine Inhibits Cancer Stem Cells

In a tumor, cancer stem cells (CSCs) are cells that can undergo continuous self-renewal and generate heterogeneous lines of cancer cells. CSCs also contribute to tumor initiation and the relapse of cancers [47,48,49]. CSCs are characterized by the deregulation of cellular energetics, promoting an inflammatory state in the tumor, evading apoptosis, avoiding immune destruction, and resisting anticancer drugs [49]. At a molecular level, Wnt/β-catenin, Hedgehog, Notch, JAK-STAT (Janus kinase/signal transducers and activators of transcription), NF-κB, PI3K/Akt/mTOR, and TGF/SMAD are fundamental signaling pathways regulating self-renewal and differentiation in CSCs [50,51]. Therefore, finding chemopreventive agents capable of targeting those signaling pathways could help prevent tumor formation. Piperine influences all these pathways directly or indirectly.

On colorectal and breast cancer cell lines, piperine has been shown to inhibit the Wnt/β-catenin signaling pathway [51], and PI3K/Akt/mTOR signaling pathways could also be influenced by piperine [32,39]. Furthermore, human cervical cancer treated with piperine and mitomycin-C resulted in inactivating STAT3/NF-κB, leading to suppression of the Bcl-2 signaling pathway [52] (Figure 4).

### 2.4. Piperine Induces Cell Cycle Arrest

The dysregulation of cell cycle control, a fundamental cellular process responsible for maintaining proper cell proliferation and safeguarding cellular integrity, is commonly associated with cancer development [53,54]. Two groups of proteins, called cyclins and cyclin-dependent kinases (CDKs), are responsible for the progression of the cell through various checkpoints.

Piperine has shown an inhibition capacity of different protein regulators and checkpoints. Fofaria et al. showed that piperine treatment inhibits the growth of melanoma cells SKMEL 28 and B16F0 in a dose- and time-dependent manner by arresting the cell cycle of both cell lines in the G1 phase [55]. Their results correlated with the down-regulation of cyclin D1 and the induction of p21 [55]. Furthermore, the phosphorylation of H2AX at Ser139 supposed that intracellular ROS formation induced DNA damage, leading to apoptosis [55]. The antiproliferative effect of piperine was also achieved in HT-29 colon carcinoma cells by causing G1 phase cell cycle arrest via the inhibition of cyclins D1 and D3 and their activating partner, cyclin-dependent kinases 4 and 6. A reduction in retinoblastoma protein phosphorylation and an up-regulation of p21/WAF1 and p27/KIP1 was also promoted by piperine [55]. Piperine can also inhibit the cell cycle at the G2/M phase in cancer cells via the down-regulation of G2-associated (cyclin B, CDK1, Cdc25C) proteins, the induction of p21, and the enhancement of the phosphorylation of both CDK1 and checkpoint kinase 2 (Chk2) [18,56,57] (Figure 4).

### 2.5. Piperine Selectively Inhibits the Growth of Cancer Cells

One of the main limitations of anticancer drugs is their frequent and severe toxic side effects, which are caused by their inability to selectively target cancer cells [58]. In an attempt to address this limitation, piperine has emerged as a promising candidate due to its selective action on cancer cells while sparing non-transformed cells in vitro. In a study by Si et al., normal ovarian cells exposed to 20 µM piperine did not show significant effects on viability. However, exposure to 8 µM piperine caused a significant decrease in ovarian cancer cell viability after 48 h [16]. Similarly, when evaluating prostate cells, piperine displayed no cytotoxicity up to 80 µM on prostate epithelial cells, in contrast to two out of the three prostate cancer cell lines tested [41]. In addition, a 200 µM dose of piperine had a weaker growth inhibitory effect on human hFOB osteoblasts after 72 h of incubation than on human osteosarcoma cells [18] (Figure 4). Nevertheless, in vivo studies are needed to confirm and better understand the selectivity of piperine for cancer cells, as these studies will provide insights into whether piperine’s preferential action against cancer cells observed in vitro can be replicated within living organisms.

### 2.6. Piperine Inhibits Cancer Invasion and Metastasis Process

Piperine inhibits cancer invasion and metastasis process by having a dual therapeutic potential in cancer by manifesting anti-angiogenic effects, inhibiting the formation of new blood vessels, and displaying anti-metastatic activity, impeding cancer cells from invading other tissues. This is explored in two distinct sections below.

#### 2.6.1. Anti-Angiogenic Effects of Piperine

Tumor progression is a step characterized by the activation of pathways that promote cell migration and the creation of further blood vessels (angiogenesis) that will allow them to self-sustain via the angiogenic factor release, such as the vascular endothelial growth factor (VEGF) [59]. In this regard, an in vivo and in vitro experimental study demonstrated that piperine inhibited tubule formation, which is a crucial step in angiogenesis [60], and inhibited angiogenic activity induced by collagen, caused by breast cancer cells. Piperine also blocked the phosphorylation of Ser 473 and Thr 308 residues of Akt involved in regulating endothelial cells and, therefore, angiogenesis [17]. To assess the effect of piperine on angiogenesis, another study evaluated VEGF expression levels in the vicinity of piperine. This study found that piperine treatment reduced VEGF expression in a dose-dependent manner, suggesting that the alkaloid negatively regulates this key growth factor in cancer cell migration [61]. The effect of piperine on crucial angiogenic factors involved in angiogenesis and tumor progression was elucidated in a study conducted in a human breast cancer cell line [61]. Piperine down-regulated MMP-9 and VEGF mRNA expressions in a dose-dependent manner. In addition, piperine treatment increased the expression levels of E-cadherin, a cell adhesion molecule required to maintain extracellular matrix integrity and cell-to-cell contact, thereby supporting the anti-metastatic potential of this alkaloid [62] (Figure 4).

#### 2.6.2. Anti-Metastatic Activity of Piperine

Metastasis is a process by which cancer cells escape and invade other tissues. Although not all cancer cells reach this stage, metastasis is the leading cause of death from cancer [63]. An experimental test on a murine lung metastasis model also demonstrated piperine’s anti-metastatic activity. Indeed, piperine effectively reduced the size of tumor nodules and the levels of uronic acid and hexosamine, both involved in the metastasis pathway. Hydroxyproline, a collagen metabolite and a potential marker of tumor cell infiltration into the bones, indicating the presence of metastasis, was also quantitatively reduced via the treatment with piperine [64]. Another research study investigating the effect of piperine on a human gastric cancer cell line showed that it inhibits the expression of IL-6, which has a prominent role in cancer cell invasion and metastasis via the c-Src/RhoA/ROCK signaling pathway [56,65]. Further experiments found that piperine inhibited tumor migration and progression by lowering the expression of MMP-13 and -9 and that high doses of piperine significantly reduced lung metastasis [63,64]. Accordingly, experimental studies have elucidated the anti-metastatic activity of piperine by regulating the pathways that involve matrix metalloproteinases [18].

**Table 1 cancers-15-05488-t001:** Piperine mechanisms and targets in in vitro and in vivo models for cancer therapy.

Targets	Model	Mechanisms	References
In vitro			
Inflammation	LPS-induced inflammation PBMCs UV-B induce oxidative stress	Inhibition of IL-2 and IFN-γ production; Inhibition of ROS/RNS production; Inhibition of the expression of proinflammatory mediators (e.g., p38, iNOS); Down-regulation of IL-1β, NF-κB, and oxidative stress-associated genes.	[26,27,28,29]
Cell death	MCF-7, HepG2, HeLa, and PC3, Bel-7402/5-FU cell lines	Induces nuclear condensation, externalization of membrane phospholipid phosphatidylserine, DNA cleavage, increases intracellular ROS, disrupts mitochondrial membrane potential, and activates caspase-3. Inhibits mTORC kinase and TrxR activity, regulates LC3, p62, and beclin-1 and induces formation of autophagosomes. Increase in intracellular Ca2+, ROS, and induction of Fenton reaction. Inhibition of STAT3.	[15,32,35,40,41,43,44,45,46]
Cancer stem cells	HCT116, SW480, DLD-1, and A549 cell lines	Inhibition of Wnt/β-catenin signaling pathway; Inhibition of PI3K/Akt/mTOR signaling pathway; Inactivation of STAT3/NF-κB; Inhibition of Bcl-2 signaling pathway.	[32,39,51,52]
Cell cycle	SKMEL 28, B16F0, and HT-29	G1 and G2 phases cell cycle arrest; Down-regulation of cyclin D1 and D3; Inhibition of CDK4 and 6; Up-regulation of p21/WAF1 and p27/KIP1; Phosphorylation of H2AX; Down-regulation of cyclin B, CDK1, and Cdc25C proteins.	[18,55,56,57]
Cancer cells growth	Hep G2, HOS, U2OS, and hFOB	Significantly decreased cancer cells viability; Did not affect normal cells;	[16,18,41]
Invasion and metastasis	MDA-MB-231, U2OS, AGS, and 143B cell lines	Inhibition of tubule formation; Down-regulation of MMP-9 and VEGF mRNA expression; Inhibition IL-6 expression; Down-regulation of expression of MMP-13 and MMP-9.	[56,60,61,62,63,64,65]
In vivo			
Inflammation	Carrageenan-induced paw edema; Formalin-induced arthritis; Croton oil-induced gran-uloma pouch; Cotton pellet-induced granuloma.	Significant inhibition of inflammation and arthritic symptoms (e.g., pain, edema); Inhibition of IL-6, MMP-13, and AP-1.	[30,31,32]
Cell death (apoptosis, autophagy)	Xenograft mouse model; Syngeneic Balb/c mice model.	Up-regulation of Bax and p53 expression; Reduction in Bcl-2 expression; Modulation of Wnt/β-catenin pathway; Synergistically inhibits cancer cell proliferation with Celecoxib.	[36,42,66]
Invasion and metastasis	Murine lung metastasis; Mice syngeneic to 4T1 cells.	Reduced uronic acid, hexosamine, and hydroxyproline levels.	[63,64]

## 3. Piperine Enlightens the Dark Side of Cancer Therapy

Today, chemotherapy and radiotherapy remain the most effective treatments for many types of cancer. Unfortunately, a proportion of cancer patients suffer from mild to serious side effects [67,68].

Researchers are making extraordinary efforts to improve cancer therapy efficacy, one of which is using natural products (supplements). In addition to enhancing cancer patients’ quality of life, many phytochemicals such as piperine can protect healthy cells, reduce multidrug resistance, increase drug penetration and its concentration inside cancer cells, and reduce the incidence of weight loss, malnutrition, and the severity of comorbidities [67].

### 3.1. Radiosensitization and UV-Phototherapy

Radiation therapy deposits high physical radiation energy on the cancer cells, leading to their destruction [69]. However, dose-limiting normal tissue toxicity, as well as the radioresistance of some tumors, limit its clinical use in cancer therapeutics [70]. At a molecular level, radiation may directly or indirectly damage DNA by producing free radicals generated when water is ionized or excited by radiation [70]. Thus, agents are needed to increase radiation-killing effectiveness and to prevent radiation-induced damage to normal cells and tissues without causing systemic toxicity. Fortunately, many chemopreventive phytochemicals could rightly fit in this approach thanks to both their cancer-preventive and therapeutic activities [71,72]. A recent study has demonstrated that piperine acts as an antioxidant in normal cells while behaving as a prooxidant in cancer cells [73]. This differential behavior enhances the radiosensitivity of cancer cells without affecting the sensitivity of normal cells. Furthermore, piperine can also increase ROS formation, leading to cell death via the dissipation of mitochondrial membrane potential [35]. Therefore, piperine can enhance ionizing radiation-induced apoptosis in cancer cells. Furthermore, in B16F10 mouse melanoma cells, the piperine combination with UVB has been reported to enhance cell death by elevating intracellular ROS formation and intracellular Ca^2+^ homeostasis impairment [24] (Figure 5).

On the other hand, UV-phototherapy kills cancer cells in the skin and its effectiveness is due to its capacity to expose only suitable skin areas. Nevertheless, UV-phototherapy can cause skin burns and increase skin cancer risk, like in sunlight-UV exposure. Many botanical agents, such as piperine, can prevent the risk of skin cancer [74]. Piperine attenuates the (UV)-R-induced DNA damage in human HaCaT keratinocytes, according to Jaisin et al. and Verma et al. [12,75]. Furthermore, piperine also inhibits UV-R-induced cell cycle arrest in HaCaT cells. UV-R alone induced p21 up-regulation, whereas, in combination with piperine, no significant up-regulation was observed [75]. Transcriptional and translational analysis revealed that piperine pretreatment suppresses the activation of NF-κB and attenuates the Bax/Bcl-2 expression in UV-R exposed keratinocyte cells, leading to the survival pathway switch [75,76,77] (Figure 5).

### 3.2. TRAIL-Based Therapy

The tumor necrosis factor-related apoptosis-inducing ligand (TRAIL), also known as Apo2L, is a type 2 membrane protein that belongs to the TNF superfamily. With the ability to kill cancer cells selectively by activating a signaling pathway involved in the innate immune system, it has become one of the few tumor-selective agents [78]. However, TRAIL therapy is limited by many cancer cells developing immunity to TRAIL and escaping destruction by the immune system [24,79]. Therefore, a study performed by Abdelhamed et al. aimed to assess the synergistic effects of 55 compounds with TRAIL on TNBC (Triple-Negative Breast Cancer) cells, including both TRAIL-sensitive and TRAIL-resistant cell lines (MDA-MB-231 and MDA-MB-468, respectively). Piperine was identified as a compound with synergistic effects with TRAIL in both cell lines, accompanied by a reduction in p65 phosphorylation and survivin expression [21].

Piperine exhibited differential cell selectivity regarding growth suppression when compared to TRAIL and its synergistic interaction with TRAIL. While the suppression of MDA-MB-468 cell growth by piperine was likely due to a cytostatic effect via G2/M cell cycle arrest, the synergistic effect with TRAIL appeared to involve direct cytotoxicity via apoptosis induction [21]. The study also indicated that the higher sensitivity of MDA-MB-468 cells to piperine was less likely to be solely due to G2/M arrest induction, as other compounds with similar cell cycle effects did not yield the same results [21]. The synergistic effect of piperine with TRAIL appeared to be linked to the suppression of p65 and survivin, which are related to TRAIL responsiveness [80,81]. Cell cycle-arrested cells are known to exhibit increased sensitivity to TRAIL, suggesting that piperine-induced G2/M arrest may contribute to both cytostatic and synergistic effects in MDA-MB-468 cells [21]. Furthermore, combination therapy involving piperine and an anti-DR5 monoclonal antibody significantly inhibited the growth of orthotopically-implanted tumors and appeared to reduce lung metastasis, showing promise in prolonging the survival rates of mice. In conclusion, the study’s findings support the potential of piperine as an adjuvant to enhance the efficacy of TRAIL-based therapy for TNBC patients [21].

### 3.3. Bioavailability of Drugs

Several factors determine a cancer drug’s bioavailability, including how the drug is released from its pharmaceutical dosage form, whether it is stable in the gastrointestinal tract, whether it dissolves easily, how quickly it passes through the gut wall, and whether it undergoes pre-systemic metabolism [82,83]. In the world of medicine, piperine is considered to be the first bioavailability enhancer that has been scientifically validated [19]. The bioavailability enhancement induced by piperine can be due to several mechanisms. For example, piperine can act in transporter proteins like ATP binding cassette (ABC) transporters, a transmembrane protein responsible for transporting a variety of substrates across extra- and intracellular membranes. ABC transporters are abundantly expressed in the epithelial layer of the gut wall and tumors and are responsible for anticancer drug efflux. Nevertheless, piperine can overcome this limitation. Studies have shown that piperine inhibits P-gp and cytochrome P450 3A4 (CYP3A4) [19]. Furthermore, piperine enhanced the bioavailability of various well-known chemopreventive natural agents (e.g., resveratrol, curcumin), by inhibiting glucuronidation, significantly increasing the plasma concentration of resveratrol when given with resveratrol [84] (Figure 6). Nevertheless, it is worth noting that human studies rarely reported effects that were considered adverse. Their suitability for detailed risk assessment is limited due to several factors. Firstly, there was an insufficient focus on safety parameters beyond drug interactions. Secondly, there was a lack of investigation into potentially adverse effects observed in animal studies. Moreover, the combined administration of piperine with other substances was not extensively explored in these human studies [85]. This leaves a gap in our understanding of the full spectrum of potential effects and safety considerations related to piperine and its interactions with other compounds.

### 3.4. Multidrug Resistance

Similar to antibiotic resistance, multidrug resistance (MDR) in cancer treatment refers to the ability of cancer cells to survive treatment with various anticancer drugs [86]. Growing evidence suggests that MDR is mediated by the increased efflux of chemotherapeutic drugs, which reduces the drug’s absorption by cancer cells [87]. Furthermore, one of the major mechanisms of MDR is the efflux of anticancer drugs by ABC transporters [88]. Oncogene mutations, tumor microenvironment (TME) changes, tumor heterogeneity, mutations at the target site, or epigenetic changes can also enhance the MDR’s development [89,90].

Phytochemicals, particularly alkaloids such as piperine, have been extensively researched for their potential to reverse MDR in cancer cells. They achieve this by affecting the activity or expression of ABC transporters, as well as by targeting additional molecular factors that can synergize with anticancer drugs [91,92]. Wojtowicz et al. discovered that piperine targets diverse drug resistance mechanisms in human ovarian cancer cell line W1 and its paclitaxel and topotecan-resistant sublines [93]. Immunofluorescence and Western blot analyses demonstrated that piperine augmented the expression of protein tyrosine phosphatase receptor type K (PTPRK). This protein was found to be down-regulated in 15 drug-resistant ovarian cancer cell lines and aldehyde dehydrogenase 1 family member A1 (ALDH1A1) positive CSCs’ population [20]. Piperine can also down-regulate P-gp and BCRP expression, limiting their ability to reject anticancer drugs [20]. Furthermore, piperine treatment down-regulates MDR1, MRP1, and BCRP genes [94,95]. Piperine can also influence the tumor microenvironment by decreasing the extracellular matrix (ECM)’s protein levels (COL3A1, TGFBI), leading to higher drug concentrations in the cell [20] (Figure 6). As a result of these combined factors, cancer cells become more susceptible to cytotoxic drugs.

## 4. Conclusions

This comprehensive review provides an in-depth exploration of the potential anticancer properties of piperine, shedding light on their mechanisms of action and therapeutic potential. The review emphasizes the significance of targeting key cellular processes involved in cancer development and progression. Piperine has exhibited remarkable preventive effects by inhibiting cell proliferation, inducing various cell deaths, and modulating signaling pathways implicated in cancer initiation. Moreover, piperine has emerged as a promising adjuvant therapy when combined with conventional anticancer drugs, as it enhances their efficacy by improving drug bioavailability, inhibiting drug efflux transporters, and modulating drug resistance pathways. The chemopreventive properties of piperine are further highlighted via its various mechanisms.

In addition to the current findings, several future perspectives warrant exploration to advance the understanding and application of piperine in cancer therapy. Further studies should aim to elucidate the specific molecular targets and pathways influenced by piperine. Thus, researchers can gain a deeper understanding of its precise mechanisms of action, allowing for the development of more precise and effective therapies. Identifying these specific targets and pathways can also pave the way for personalized treatment strategies, optimizing therapeutic benefits while minimizing adverse effects.

Furthermore, the investigation of piperine’s potential synergistic effects with other compounds is of paramount importance. Uncovering novel combination therapies with enhanced efficacy could lead to more potent and individualized treatment options. These combinations have the potential to increase response rates and significantly impact patient outcomes, making them a crucial area of study.

Optimizing the delivery systems for piperine administration represents a promising avenue of research. Enhancing targeting and bioavailability will ensure that piperine reaches its intended site of action within the body, maximizing its therapeutic benefits [96]. This optimization could lead to improved treatment outcomes and a higher quality of life for patients.

Conducting well-designed clinical trials is critical to assess the safety, tolerability, and effectiveness of piperine-based therapies in diverse patient populations. These trials provide valuable data on the real-world applicability of piperine in cancer therapy. If proven effective, piperine-based treatments may become a standard part of cancer care, offering patients a broader range of treatment options and potentially increasing their chances of survival. Moreover, exploring the chemopreventive effects of piperine in high-risk populations holds significant promise. This research can provide valuable insights into piperine’s potential as a preventive agent against the development of cancer. By identifying individuals at a higher risk of cancer and providing them with preventive strategies involving piperine, we can significantly reduce cancer incidence. This approach benefits individuals and society alike by reducing the societal and economic burden of cancer. These ongoing research efforts and future directions will contribute to the advancement of piperine-based therapies, paving the way for their integration into the clinical management of cancer. Ultimately, the comprehensive understanding and optimized utilization of piperine hold tremendous potential for improving patient outcomes and alleviating the burden of this devastating disease on individuals and society.

## Figures and Tables

**Figure 1 cancers-15-05488-f001:**
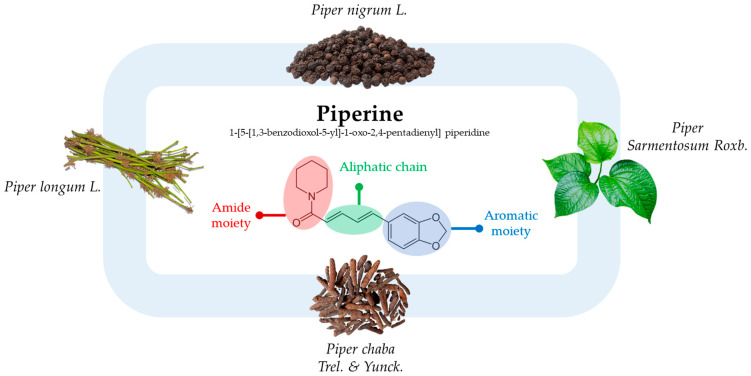
Piperine’s chemical structure and some plant sources.

**Figure 2 cancers-15-05488-f002:**
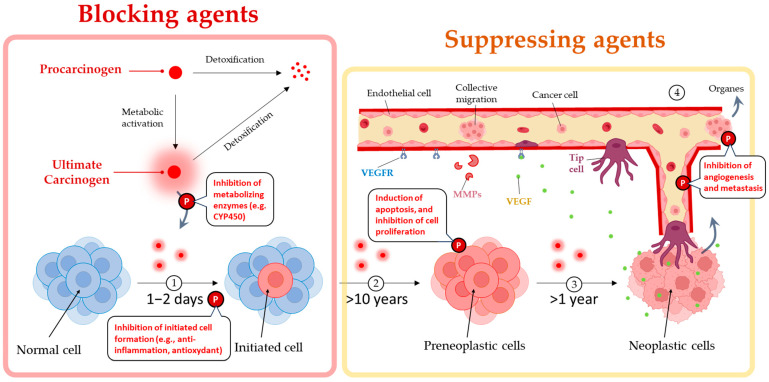
Chemopreventive blocking and suppressing agents interfere with the metabolic conversion of procarcinogens to their ultimate intermediates, where the key steps in carcinogenesis were 1. Initiation, referring to the transformation of a normal cell into an initiated cell, 2. Promotion, leading to the formation of preneoplastic cells, 3. Progression, advancing to neoplastic cells, and finally, 4. Angiogenesis and Metastasis, where VEGFR promotes the formation of new blood vessels and the migration of cancer cells to other organs. At the same time, MMPs facilitate these processes by remodeling tissues and breaking down barriers. The pleiotropic effects of piperine “P” are also briefly illustrated.

**Figure 3 cancers-15-05488-f003:**
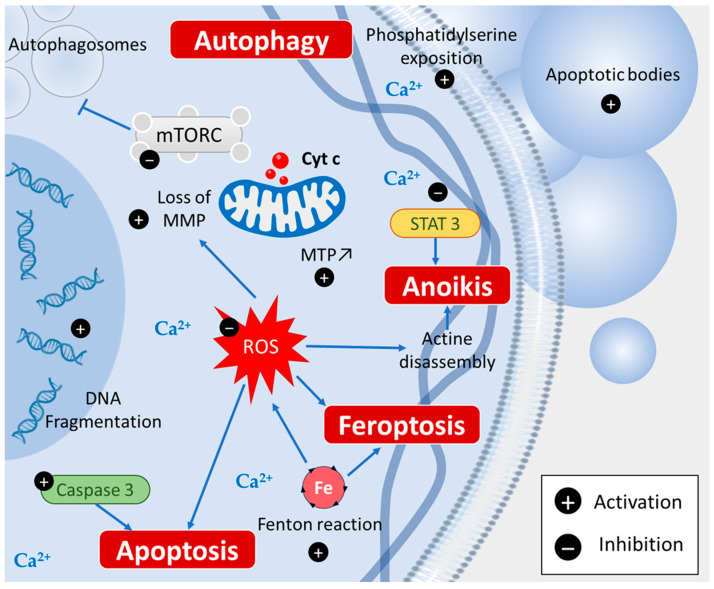
Piperine induces various types of cell death: apoptosis via caspase 3 activation, ferroptosis via Fenton reaction, autophagy by inhibiting mTORC signaling, and anoikis via STAT3 inhibition.

**Figure 4 cancers-15-05488-f004:**
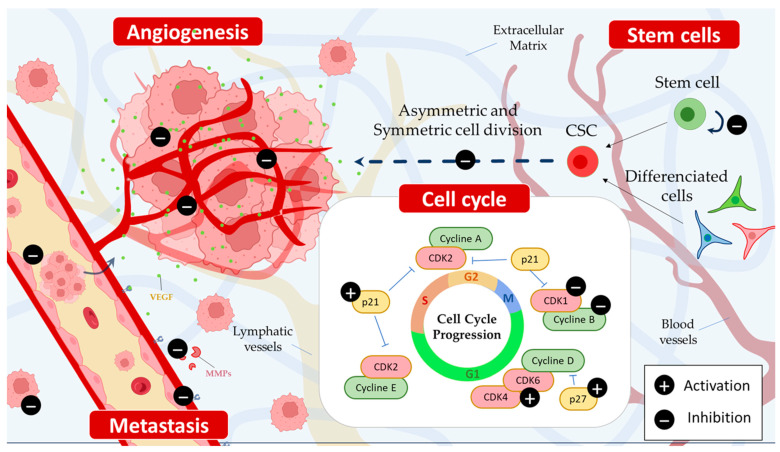
Piperine inhibits angiogenesis and metastasis by reducing the expression of vascular endothelial growth factor (VEGF) and down-regulating matrix metalloproteinase (MMP). Additionally, it hinders the self-renewal capacity of cancer stem cells (CSCs) via both direct and indirect mechanisms, targeting critical pathways such as Wnt/β-catenin, Hedgehog, and Notch. Moreover, piperine induces cell cycle arrest at various phases, including G1, G1/S, or G2/M.

**Figure 5 cancers-15-05488-f005:**
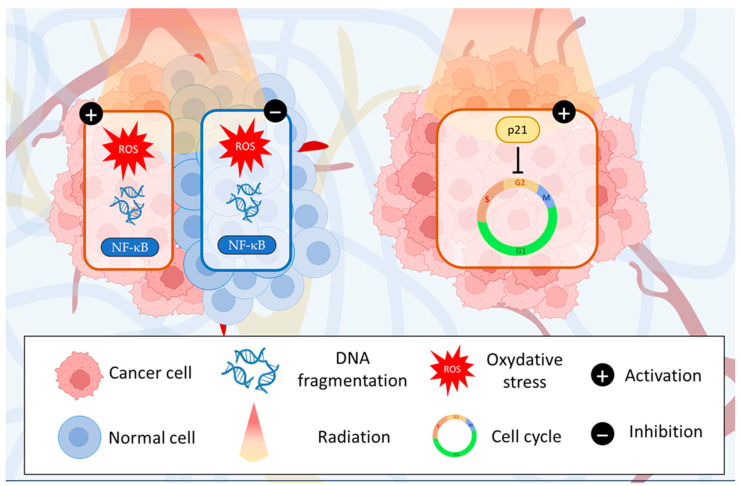
Enhancing radiation therapy while minimizing side effects: piperine’s impact on oxidative stress, DNA damage, NF-κB activation, and cell cycle regulation in cancer and normal cells.

**Figure 6 cancers-15-05488-f006:**
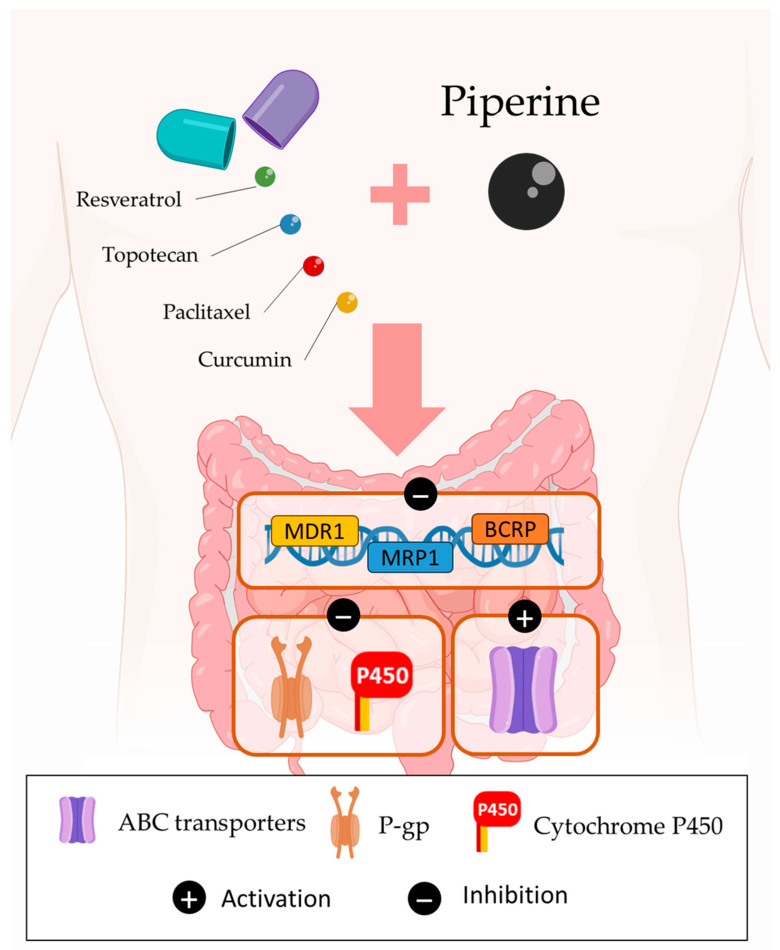
Optimizing anticancer drug bioavailability with piperine: inhibition of the up-regulation of genes (e.g., MDR1, MRP1, BCRP) and cellular proteins (e.g., P-gp and cytochrome P450) and activation of ABC Transporters.

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
