# Peer review of "The Promise of Piperine in Cancer Chemoprevention"

_cancers, 2023, doi:10.3390/cancers15225488_

Round 1

Reviewer 1 Report

Comments and Suggestions for Authors

In this article, authors Benayad et al. provided a comprehensive review of anti-cancer effects of piperine, a natural compound, that is extracted from the piper plants. The article specifically reviews the anti-tumor effects of this compound and its mechanism of action in various organ site cancers. This is a well written review with clarity. However, addressing some of the concerns as noted below would strengthen it.

1.       Several studies showing anti-cancer effects with this compound have shown that it will work through various mechanisms. It looks like the compound has pleotropic effects showing effects on multiple targets. Authors should comment on this aspect.

2.       Although the title and abstract highlights chemoprevention, most of the studies cited were either in vitro or those performed in therapeutic settings. Authors must emphasize a little more on chemopreventive effects of piperine.

3.       Authors focus was primarily on in vitro cell lines studies. The addition of in vivo data in animal models will be helpful. Please add two tables, one for in vitro studies and the other for in vivo chemoprevention studies.

4.       Did any of the studies reviewed evaluate the safety and toxicity profiles of the compound?

5.       Any information on clinical trials (possibly chemoprevention) would be helpful.

Reviewer 2 Report

Comments and Suggestions for Authors

The research article by Salma Benayad1 et al. presented the work entitled "The Promise of Piperine in Cancer Chemoprevention." The manuscript explores how piperine can inhibit molecular events and signaling pathways associated with cancer development, making it a promising tool in cancer chemoprevention. Additionally, the manuscript mentions that piperine may enhance the effectiveness of conventional cancer treatments, such as UV-phototherapy and TRAIL-based therapy, offering more robust cancer management strategies. Overall, the manuscript provides a clear introduction and sets the stage for further discussion. By addressing the comments given below for improvement, the research article can become more coherent, informative, and engaging for readers.

  1. Opening Sentence: The opening sentence effectively highlights the severity of the global cancer crisis, but it might be beneficial to add some context or statistics to convey the scale of the problem more explicitly. For instance, you could provide statistics on the increasing cancer burden over the years.
  2. Unique Property of Piperine: The unique property of selectively targeting malignant cells is a crucial point and well-placed in the introduction. However, you could emphasize why this is important in the context of cancer therapy, perhaps by briefly mentioning the challenges of distinguishing malignant cells from healthy ones in treatment.
  3. Versatility in the Battle Against Cancer: The introduction concludes well, emphasizing piperine's versatility. Consider reinforcing this point by briefly mentioning the various cancer types it can target.
  4. Clarity and Flow: Ensure that the text flows smoothly from one point to the next. Each paragraph should naturally lead to the next without abrupt transitions.
  5. Subsection Organization: The introduction mentions "blocking agents" and "suppressing agents" but does not immediately explain how piperine fits into these categories. Consider making a smoother transition into the subsection about piperine's specific mechanisms.
  6. Formatting and Organization: Consider formatting this subsection into smaller paragraphs or sections to improve readability. Each type of cell death (apoptosis, autophagy, ferroptosis, and anoikis) could be presented in a separate section with clear headings.
  7. Grammar and Clarity: In section 2.2, some sentences are quite long and complex. Simplify and break them down for clarity and easier comprehension. 
  8. Detail and Significance: After discussing the various ways piperine induces cell death, consider briefly summarizing these findings' significance in cancer prevention and therapy.
  9. 2.4. Pipepine induces cell cycle arrest. Be consistent in formatting: "Pipepine" should be "Piperine."
  10. Use concise language and structure sentences for better clarity. For example, "A recent study has shown that piperine can act as an antioxidant in normal cells but act as a prooxidant in cancer cells, enhancing the radiosensitivity of cancer cells without influencing the sensitivity of normal cells." can be rephrased for clarity. 
  11. Discuss the study's specific findings by Abdelhamed et al., providing more context on how piperine synergistically enhances TRAIL therapy. 
  12. Consider giving specific examples or studies demonstrating piperine's impact on bioavailability, such as resveratrol and curcumin.
  13. Begin the conclusion by summarizing the key findings and contributions of the review.
  14. Provide more context for the potential benefits of these future research directions and how they might impact cancer therapy and patient outcomes.
  15. Address the need for further studies to explore the safety and efficacy of piperine-based therapies in a clinical context.

Reviewer 3 Report

Comments and Suggestions for Authors

Thank you very much for allowing me to review the review article entitled "The Promise of Piperine in Cancer Chemoprevention" (cancers-2689302), this review is preesented to Section "Cancer Epidemiology and Prevention" in the Special Issue "Chemoprevention Advances in Cancer".
First, the presented review is in line with Section and the Special Issue.
The aim of this review is to explore piperine's potential in cancer prevention and treatment, suggesting it could be a valuable tool in managing cancer.

This is a comprehensive review, I want to congratulate you for the graphics that are very informative and facilitate the understanding of the subject, being complementary to the text.
However, I consider that it is necessary to incorporate a section on methodology, reviews are currently a type of article that allows us to integrate knowledge and professionals to keep up to date on issues that we would need to read many other articles. However, this implies that the period being reviewed, the source of data used, and the methodology used in the review must be indicated. These aspects are essential to connect different reviews on the same topic and to establish a chronology of scientific knowledge, so I suggest that the authors incorporate this section.
The abstract should be more structured.

Round 2

Reviewer 1 Report

Comments and Suggestions for Authors

Authors have addressed the comments satisfactorily.

Author Response

The authors thank the reviewer for the time dedicated to the revision of our manuscript and the thoughtful comments.

Reviewer 2 Report

Comments and Suggestions for Authors

Subsection 2.1: Consider adding a concise concluding statement that synthesizes the findings. This could reinforce the idea that piperine consistently demonstrates anti-inflammatory effects across various models, making it a promising candidate for further exploration in inflammatory-related conditions.

2. Consider adding subheadings within the section to delineate the discussion into subsections clearly. For example, you can have subsections like Anti-Angiogenic Effects of Piperine and  Anti-Metastatic Activity of Piperine.

3. In the sentence "This leads to a blocking of the adhesion of neutrophils to endothelium in a time- and concentration-dependent manner" you can consider rephrasing for clarity "This leads to the blocking of neutrophil adhesion to the endothelium in a time- and concentration-dependent manner"

4. In the sentence "Combining all these factors makes cancer cells more susceptible to cytotoxic drugs" consider providing a smooth transition or introductory phrase to enhance the flow. For example, "As a result of these combined factors, cancer cells become more susceptible to cytotoxic drugs."

5. Consider breaking down longer sentences into shorter ones for improved clarity. For example, the sentence starting with "A study performed by Wojtowicz et al..." is quite lengthy. Breaking it down can enhance readability.

6. Ensure subject-verb agreement. In the sentence "Furthermore, piperine down-regulated MMP-9 and 281 VEGF mRNA expression in a dose-dependent manner" it should be "down-regulated MMP-9 and VEGF mRNA expressions" to maintain agreement.

Author Response

"Please see the attachment".

Reviewer 3 Report

Comments and Suggestions for Authors

I have meticulously reviewed the revised version of the review article entitled "The Promise of Piperine in Cancer Chemoprevention" (cancers-2689302), along with the authors' responses to the previously raised comments.

I am notably impressed with the enhancements in the manuscript, which now presents a more specified and substantiated review.

Regarding my request for a clearer explanation of methodological aspects, the authors have addressed these concerns between lines 86 and 99.

Furthermore, the authors have augmented the quality of the abstract by indicating the necessity for future research, though they have omitted to include the time period that was reviewed to conduct this assessment.

On the whole, I consider that the article exhibits a significant enhancement in the quality of the review carried out.

Author Response

The authors would like to express their sincere gratitude to the reviewer for the valuable feedback and thoughtful insights, which significantly contributed to the improvement of the manuscript.

Regarding the time period, it was included in the revised version of the manuscript in the following sentence: 

"Our focus was primarily on scrutinizing studies published between 2002 and 2023, with the inclusion of two earlier studies for contextual purposes."